# Personalized Diagnosis and Treatment for Neuroimaging in Depressive Disorders

**DOI:** 10.3390/jpm12091403

**Published:** 2022-08-29

**Authors:** Jongha Lee, Suhyuk Chi, Moon-Soo Lee

**Affiliations:** 1Department of Psychiatry, Korea University Ansan Hospital, Ansan 15355, Korea; 2Department of Psychiatry, Korea University Guro Hospital, Seoul 08308, Korea; 3Department of Life Sciences, Korea University, Seoul 02841, Korea

**Keywords:** major depressive disorder, resting-state functional connectivity, machine learning, classification, neuroimaging, diagnosis, prediction

## Abstract

Depressive disorders are highly heterogeneous in nature. Previous studies have not been useful for the clinical diagnosis and prediction of outcomes of major depressive disorder (MDD) at the individual level, although they provide many meaningful insights. To make inferences beyond group-level analyses, machine learning (ML) techniques can be used for the diagnosis of subtypes of MDD and the prediction of treatment responses. We searched PubMed for relevant studies published until December 2021 that included depressive disorders and applied ML algorithms in neuroimaging fields for depressive disorders. We divided these studies into two sections, namely diagnosis and treatment outcomes, for the application of prediction using ML. Structural and functional magnetic resonance imaging studies using ML algorithms were included. Thirty studies were summarized for the prediction of an MDD diagnosis. In addition, 19 studies on the prediction of treatment outcomes for MDD were reviewed. We summarized and discussed the results of previous studies. For future research results to be useful in clinical practice, ML enabling individual inferences is important. At the same time, there are important challenges to be addressed in the future.

## 1. Introduction

Psychiatric disorders such as depressive disorders are highly heterogeneous. Depressed mood and markedly diminished interest in previously enjoyed activities are key characteristics of major depressive disorder (MDD), but symptoms show numerous heterogeneous constellations. This indicates that the characteristics of MDD are not confined to certain single parameters but may associate with multiple bio–psycho–social dimensions. Thus, there is a need to integrate various data forms for better knowledge on the disease. Many past studies have identified a range of possible biomarkers for depressive disorders. However, these results have not yet been successfully integrated into clinical practice for diagnosis and treatment. In addition, there is heterogeneity in the clinical presentation of MDD and its responses to treatment. Accordingly, identifying applicable biomarkers that can predict diagnosis and treatment outcomes would be useful in clinical practice. Currently, depressive disorders are diagnosed by trained professionals using their clinical judgement. However, this process is time-consuming and depends on the subjective judgement of the clinician. In clinical practice, it is common not to receive additional help from brain imaging, although brain imaging provides a noninvasive evaluation of brain structure and function and provides a deeper understanding of the neuropathophysiology of MDD. Brain imaging studies should be introduced in the clinical management of depression. In this process, both objective biomarkers and subjective impressions of clinicians are required for accurate classification and treatment decisions. There is a need for objective biomarkers for more effective diagnosis and treatment of depression. From this perspective, we would like to review the status of brain imaging studies by dividing them into two large groups: diagnostic biomarkers and prediction of treatment outcomes. For a more specific search, we focused on MDD, the most representative disease in depression.

Currently, there are many neuroimaging methodologies available. For example, magnetic resonance imaging (MRI) comprises many methods, including structural MRI (sMRI) and functional MRI (fMRI). sMRI has been conventionally used in patients with clinical depression, whereas the fMRI technique can provide other kinds of useful information, including brain functional connectivity, which provides useful insights into the pathophysiology of depression in relation to functional connectivity in the prefrontal–limbic and prefrontal–striatum systems in patients with MDD. However, such studies are not useful for the clinical diagnosis and prediction of MDD outcomes at an individual level [1]. There are also many clinical studies using neuroimaging in MDD, and most of them reported differences between patient and control populations, but clinicians need to make inferences at an individual level in most clinical settings [2]. The use of machine learning (ML) techniques is a possible way to make inferences beyond population-level analyses, both for the diagnosis of more specific subtypes of MDD and the prediction of treatment responses [3]. ML algorithms are powerful analytical tools that enable us to integrate neuroimaging and non-imaging data, helping to make decisions on the diagnosis and treatment outcomes of individual patients. ML can generalize patterns from the input data to generate a classification based on new data. Recently, deep learning, a particular branch of ML, has been increasingly used as it can even more effectively integrate neuroimaging data and non-imaging multimodal data [4].

In this paper, we summarize the regular and distinct findings of ML in neuroimaging studies of MDD (mainly focused on MRI) among mood disorders. We comprehensively reviewed the application of ML algorithms for neuroimaging in depressive disorders. A bibliographic search of PubMed and Google Scholar was conducted in December 2021. We searched PubMed through December 2021 for relevant studies that included depressive disorders and applied an ML algorithm in neuroimaging fields for depressive disorders. We divided these into diagnosis and treatment outcomes for the purpose of prediction using ML. In addition, we attempted to classify the studies according to the characteristics of the methods used, namely structural and functional studies. By doing this, we were able to follow the existing research results based on ML algorithms more intuitively.

## 2. Considerations for ML in Neuroimaging in Depressive Disorders: Diagnosis (Table 1)

### 2.1. Structural Characteristics for the Assessment of Depression

#### 2.1.1. Structural Neuroimaging Studies for Diagnosis

Several MRI scan sequences have been used in this study, among which high-resolution T1-weighted imaging confirms gray matter thickness in volume and changes in brain morphology. Previous sMRI studies have suggested variable results of structural changes in a depressed patient group compared with a normal healthy group [35,36,37,38,39,40,41,42,43,44,45]. Thus, sMRI may be the most feasible method for clinical practice. Conventional structural neuroimaging studies have mainly focused on regional volume alterations in gray matter. However, this is insufficient, as morphometric alterations also include changes in shapes and geometric features. Differences in cortical thickness, gray matter volume, and white matter integrity were investigated. Alterations in the cortical thickness have been suggested in several brain regions. Some studies have reported that in patients with MDD, cortical thickness increases in the orbitofrontal cortex [37,38], superior frontal gyrus [36], cingulate cortex [37,39,40], and occipital cortex [40]. Other studies have reported decreased cortical thickness in the orbitofrontal cortex [41,42], insular cortex [43,44], bilateral fusiform gyrus [39], and left occipital area [45]. In a recent meta-analysis of medication-free patients with MDD, Li et al. showed a complex pattern of increased cortical thickness in the posterior cingulate cortex, ventromedial prefrontal cortex, and anterior cingulate cortex and decreased cortical thickness in the gyrus rectus, orbital segment of the superior frontal gyrus, and middle temporal gyrus [46]. 

Diffusion tensor imaging (DTI) has been used to investigate white matter connectivity and abnormalities in the brain [47]. Using eigenvalues and eigenvectors of water molecules in the brain white matter obtained by MRI, fractional anisotropy (FA), mean diffusivity, axial diffusivity, and radial diffusivity were calculated, and alterations in white matter were confirmed through changes in these values [48]. Previous DTI studies showed relatively consistent results, in that the MDD group had lower FA values than the healthy group in brain regions, including the uncinated fasciculus (UF) [49,50], superior longitudinal fasciculus [51,52], anterior limb of the internal capsule [53,54], corpus callosum (CC) [55,56,57], and inferior fronto-occipital fasciculus [50,56]. In a recent meta-analysis, adolescents and young adults with MDD showed lower FA values in the CC and frontal-subcortical circuits, which may contribute to the pathogenesis of MDD [58]. Decreased FA values in patients with MDD were associated with the severity of depressive symptoms and duration of illness [53,54,59]. Zhu et al. reported that FA values in the left anterior limb of the internal capsule were negatively correlated with the severity of depressive symptoms [53]. Longer illness duration, the number of previous depressive episodes, and treatment response were related to lower FA values [54,59,60]. The reduction of FA values in treatment-resistant/chronic MMD was significant when compared to that in first-episode MDD and healthy controls [59]. Zheng et al. reported that reduced FA values of the UF in MDD patients returned to normal FA values in healthy controls after 8-week antidepressant treatment [60]. Similarly, reduced white matter connectivity in the anterior cingulum and CC may represent a biomarker of risk for developing MDD [61], and an alteration in white matter microarchitecture has been suggested as a predictor of the treatment outcome in MDD [62,63]. 

Existing neuroimaging studies have confirmed brain changes in patients with MDD; however, these results have not been applied in current clinical practice. Most of the studies were comparative studies of MDD and healthy control groups, and there was a limitation in investigation of individual-level comparisons. ML has been presented as a method to compensate for these limitations. 

#### 2.1.2. ML in Structural Studies for Diagnosing MDD

In this section, we introduce the performance, including accuracy, sensitivity (true positive rate), and specificity (true negative rate), of ML models used in previous sMRI studies for diagnosing MDD. ML studies for diagnosing and predicting the onset of MDD have been steadily increasing, and among them, studies comparing MDD and healthy control groups have been most actively conducted [3]. Foland-Ross et al. reported that baseline cortical thickness predicted the first-onset of MDD with an overall accuracy of 70% in a five-year follow-up study on adolescent girls aged 10–15 years [5]. Lower baseline thickness of the right medial orbitofrontal cortex and thicker left insula were associated with a higher risk of developing MDD. In a ML diagnostic study, medication-naïve adolescents with first-onset MDD showed increased thickness of the superior segment of the circular sulcus of the insula compared to the healthy control group [6]. In this study, the support vector machine (SVM) method yielded the highest performance with an accuracy of 94.4% (sensitivity, 92.6% and specificity, 96.3%). ML studies using sMRI to diagnose MDD in adults with MDD have also been reported. Qiu et al. found that alteration of the cortical thickness in the right hemisphere could differentiate first-onset MDD patients from healthy controls, providing an accuracy of 78% [7]. They suggested that morphological alterations in the right hemisphere were more evident than those in the left hemisphere in diagnosing MDD. An ML study for diagnosing MDD using DTI data from 29 MDD patients and 30 healthy controls showed an accuracy of 83.05% [8]. In this study, Qin et al. showed that frontoparietal network dysfunction was associated with adult MDD and suggested alterations in this area as a diagnostic measure for MDD. A previous ML study comparing MDD patients and control groups reported that combinations of multimodal imaging and non-imaging measures may help predict late-life depression diagnoses [9]. A learning method called “alternating decision tree” showed the highest accuracy (87.27%) in predicting the diagnosis of late-life depression; poor cognitive ability and whole brain atrophy were found to be associated with late-life depression. 

Depression severity was predicted by gray matter volume in patients with bipolar and unipolar depression. Depressive severity was predicted based on the gray matter volume of the bilateral insula, but hypomanic symptom severity was not able to distinguish between unipolar and bipolar depression [10]. In contrast to the previous result that insula volume was smaller in patients with MDD than in healthy controls [43], increased volume was associated with higher symptom severity in mood disorders. These results are likely due to the influence of the bipolar disorder group among participants. In a study comparing bipolar disorder, MDD, and healthy controls, larger volumes of subcortical regions were found in the bipolar disorder group, suggesting potentially varying neuropathological processes in these two conditions [11]. In ML using DTI data, the diagnoses of bipolar disorder and MDD were predicted at the individual level [12]. The FA tract profile of the left anterior thalamic radiation was used to discriminate between bipolar disorder and MDD with an accuracy of 68.33%. These results suggest that the effects of MDD and bipolar disorder on brain structural abnormalities are different, and the accuracy of diagnostic prediction can be improved through a better understanding of the neuropathophysiology. 

ML studies for the diagnosis of depression are rapidly increasing, but most studies have a limitation of a small sample size [5,6,7,64]. In a comparative study of bipolar disorder and MDD, it was difficult to identify the characteristics of bipolar I and II because the bipolar disorder subtypes were not classified in the bipolar disorder patient group. Since this was a cross-sectional study, there is a limit to predicting future changes in bipolar disorder or the onset of comorbid psychiatric disorders in the MDD patient group. Therefore, follow-up studies are required, and when comparing the unipolar depression group with the group that changed from depression to bipolar disorder, a better understanding of the brain structural alterations of the two disorders may be possible. 

### 2.2. Functional Characteristics for the Assessment of Depression

Functional neuroimaging refers to the use of neuroimaging technologies to measure brain function. It is different from structural imaging techniques as it uses various ways to examine the activation and interaction of and between brain regions. Commonly used methods include positron emission tomography, single-photon emission computed tomography, and functional ultrasound imaging; however, the most widely used method is fMRI. Therefore, we mainly focused on recent fMRI studies on depressive disorders that incorporated ML for enhanced results. The use of ML in fMRI studies dates to the late 2000s. Fu et al. applied the SVM method to fMRI data of depressed patients and healthy controls during facial recognition tasks [13]. The authors hypothesized that there would be stronger contributions from regions that process facial expressions, such as the lateral temporal cortex, amygdala, and visual-processing networks. The ML process resulted in 74% of the patient group and 63% of the control group being correctly classified, yielding an accuracy of 68%. Post-treatment analysis resulted in 75% of partial responders and 62% of full responders being classified correctly. Cao et al. investigated resting-state functional connectivity (rsFC) in 39 MDD patients and 37 matched healthy controls [14]. The altered functional connections were identified and applied to the SVM classification, resulting in an accuracy of 84%. The modules with the highest contribution were the inferior orbitofrontal gyrus, supramarginal gyrus, inferior parietal lobule-posterior cingulated gyrus, and middle temporal gyrus-inferior temporal gyrus.

A study by Mourao-Miranda et al. also investigated the response to sad faces in depressed patients using an SVM [15]. The brain patterns of healthy controls while responding to the stimulus were analyzed, and the patterns of patients with depression were hypothesized to be outliers when compared to controls. Of the patients, 52% were correctly identified as outliers, and 79% of controls were detected as non-outliers. Additional analyses revealed that only 30% of outlier patients responded to antidepressant treatment, whereas 89% of non-outlier patients showed a response.

Zeng et al. analyzed resting-state functional connectivity in MDD patients [16]. Consensus functional connections from previous literature were identified, and many were diminished in the patients. Discriminative power was calculated using the SVM method, and the results showed that the amygdala exhibited the highest discriminative power, showing altered connectivity between the prefrontal lobe, visual cortex, cerebellum, and other limbic areas.

Many non-ML studies have shown that functional connectivity changes are inconsistent in patients with MDD. Guo et al. examined voxel-mirrored homotopic connectivity (VMHC) alterations to obtain more consistent results [17]. Two individual samples of 59 MDD patients and 31 controls and 29 MDD patients and 24 controls were included in the fMRI data acquisition. VMHC was computed using REST software. The overlap of brain clusters showing significant differences between patients and controls was generated using a mask. LIBSVM (A Library for Support Vector Machines) software (http://www.csie.ntu.edu.tw/~cjlin/libsvm/), an integrated software for support vector classification, regression, and distribution estimation, was then used to identify the prediction model. The results showed that the VMHC values in the posterior cingulate cortex and cuneus were able to differentiate MDD patients with an accuracy of 92.22% and 90.57% in each sample, respectively.

Wei et al. concentrated on the “long-term memory” of the temporal dynamics of brain activity [18]. The Hurst exponent has been reported to describe brain activity well in terms of scale-free dynamics. SVM studies involving the Hurst exponent of brain networks of 20 MDD patients and 20 healthy controls revealed a successful discrimination with an accuracy of 90%. The results showed that the right frontoparietal and default mode networks had deficits (lower memory), whereas the left frontoparietal, ventromedial prefrontal, and salience networks were excess networks (longer memory) in patients with MDD.

He et al. examined the role of microRNA-9 in the link between childhood maltreatment and MDD [19]. MicroRNA-9 is thought to be a neural substrate for childhood maltreatment. Forty patients with MDD and 34 healthy controls completed laboratory tests and underwent fMRI, resulting in higher microRNA-9 levels in patients with MDD. SVM models integrating microRNA-9 levels, childhood maltreatment severity, and intrinsic amygdala functional connectivity showed an accuracy of 85.1% in differentiating MDD patients.

Ramasubbu et al. investigated the possible effect of severity on the accuracy of machine-learned classifications [20]. Patients with MDD were divided into groups based on their Hamilton Depression Rating Scale (HRDS) scores, which were classified as mild to moderate, severe, and very severe. fMRI data from 45 patients and 19 controls were collected during the resting state and during an emotional-face matching task. Linear SVM classifiers were used to distinguish patients from controls. The very severe depression group showed an accuracy of 66%, the mild to moderate group showed an accuracy of 58%, and the severe group showed an accuracy of only 52%. The authors suggested that machine-learned patient classification using fMRI data might be limited to less severe depression.

Ramasubbu also examined patients with MDD using arterial spin labeling (ASL) MRI [21]. Of the 22 MDD patients and 22 healthy controls who underwent pseudo-continuous 3D-ASL imaging to determine regional cerebral blood flow, which was then used in combination with sex and age as SVM classifiers for detecting patients, the resulting classification had an accuracy of 77.3%, with the highest contributing features being sex and the cerebral blood flow in cortical, limbic, and paralimbic regions.

Yamasita et al. acknowledged the difficulty of neuroimaging studies owing to the differences between various study sites and their fMRI products [22]. They used a harmonization method to remove such differences in a dataset of 713 participants from four imaging sites (564 healthy controls and 149 MDD patients). The dataset was then analyzed using an ML algorithm called the least absolute shrinkage and selection operator. It was shown that the functional “under” connectivity (more negative) between the right and left insula was the largest difference between MDD patients and healthy controls. A total of 25 functional connectivities were identified for classifying MDD, of which 19 were more negative and six were more positive than healthy controls. These classifiers were used on another dataset with 521 participants from five imaging sites (264 healthy controls and 185 MDD patients) for validation, resulting in a diagnostic accuracy of 70%.

Nouretdinov et al. applied the transductive conformal predictor (TCP) method to MRI, which generated confidence measures for imaging-based predictions [23]. In fact, this study validated the accuracy of the TCP method compared to more conventional methods such as the SVM. Using sad face recognition as a predictor, the authors found diagnostic and prognostic accuracies comparable to those of the conventional methods. Patients reacted more sensitively to sad faces, and such sensitive individuals tended to respond worse to treatment, in line with the findings of previous studies.

Hahn et al. conducted a study analyzing probable diagnostic biomarkers using Gaussian process classifiers (GPC) [24]. Of the 15 conditions used as classifiers, eight were revealed to be significantly accurate in correctly identifying patients with a median accuracy of 60%: sad face, happy face, anxious face, neutral face, anticipation of no reward, anticipation of large reward, anticipation of no loss, and avoiding small loss. GPC showed a higher accuracy than the conventional SVM method in most cases. The authors also stated that a decision tree algorithm led to an accuracy of 83%, which is an improvement of 11% compared with the best GPC (anticipation of no loss).

Rosa et al. used the same dataset as Fu et al. and Hahn et al. to expand on the research [13,24,25]. The authors used a connectivity-based framework to classify the existing data. These analyses resulted in higher accuracies than the original studies (77% vs. 79% for Fu et al., and 70% vs. 85% for Hahn et al.). It should be noted that the sensitivity and specificity were lower than those in the original reports, thus emphasizing that the new framework might not necessarily be superior.

A multicenter analysis by Shi et al. used a multivariable regression algorithm named relevance vector regression to identify sleep-related MRI indicators in patients with MDD [26]. The analysis of 92 patients with MDD identified 50 MRI features distributed through the subcortical system and frontoparietal and visual networks that showed abnormal metabolism. These findings were validated using a multicenter dataset of 460 patients and 470 controls, indicating that sleep disturbance-related MRI features may be possible biomarkers of MDD.

Guo et al. suggested that traditional methods for processing functional connectivity data are highly limited in interpretation and thus proposed a novel high-order minimum spanning tree network for better analysis [27]. The results showed a classification accuracy of up to 97.54% when comparing MDD patients with healthy controls.

Sato et al. used an ML algorithm to assess not the depression itself but vulnerability to it [28]. Subjects were selected from past depression patients who had remitted at least 1 year previously. Functional connectivity was assessed while the subjects and controls looked at statements about social and moral values. They were later asked to describe their feelings as guilt, disgust, shame, or anger toward themselves or others. A specific ML algorithm, the Maximum Entropy Linear Discriminant analysis, showed that guilt-related functional connectivity changes in the anterior temporal lobe area discriminated previous depression patients from healthy controls with an accuracy of 78.26%, suggesting it as a possible biomarker of patients’ vulnerability to depression.

Han et al. examined the so-called triple network of the brain, consisting of the default mode, salience, and central executive network, to distinguish schizophrenia patients from MDD patients [29]. Twenty-one schizophrenia patients and 25 MDD patients were assessed using sMRI and fMRI, and the data were processed using supervised convex nonnegative matrix factorization. This approach was proposed to extract low-rank network patterns in latent space. The middle cingulate cortex, inferior parietal lobule, and cingulate cortex were the most discriminative between the two disorders in terms of functional connectivity, with an accuracy of 82.6%.

Yu et al. also investigated differences in functional connectivity between MDD and schizophrenia [30]. Thirty-two patients with schizophrenia, 19 patients with MDD, and 38 controls underwent fMRI scans, with the results analyzed using an SVM with intrinsic discriminant analysis. Both groups showed altered connections in the medial prefrontal cortex, anterior cingulate cortex, thalamus, hippocampus, and cerebellum. However, the groups also showed differences in the prefrontal cortex, amygdala, and temporal poles. Patient discrimination achieved an accuracy of 80.9% (84.2% for MDD, 81.3% for schizophrenia, and 78.9% for controls). The connections with the highest discriminative powers were found within the default mode network and cerebellum.

Grotegerd et al. attempted to discriminate between unipolar and bipolar depression using an fMRI pattern classification [31]. Twenty participants (10 bipolar and 10 unipolar) were asked to look at happy, negative, and neutral emotional faces during fMRI scans. The contrasts between negative and happy versus neutral faces were used as classifiers. Both the SVM and GPC algorithms were used for classification. SVM classification showed that the happy versus neutral contrast reached an accuracy of 90%, and the negative versus neutral contrast reached an accuracy of 75% for discriminating unipolar from bipolar depression. GPC classification on the other hand showed both happy versus neutral and negative versus neutral as achieving an accuracy of 70%.

He et al. examined the possibility of predicting the specific characteristics of patients with MDD using fMRI [32]. Sixty-three MDD patients and 63 matched controls underwent rsFC imaging. Their trait characteristics were measured using the Affective Neuroscience Personality Scale (ANPS), and state anhedonia was measured using the Snaith–Hamilton Pleasure Scale. SVM regression was used to predict trait and state characteristics based on changes in rsFC. Abnormal connectivity between the left amygdala/hippocampus and right amygdala/hippocampus predicted sadness scores of the ANPS, while connectivity between the medial prefrontal cortex/anterior cingulate gyrus and amygdala/parahippocampal gyrus predicted a state of anhedonia.

Not all ML approaches yield valuable positive outcomes. Maglanoc et al. implemented a ML approach to assess the relationships between clinical variables and structural and functional brain components [33]. Overall, the models showed low predictive values for depression and anxiety symptoms.

Sundermann et al. suggested that most ML approaches using fMRI results as classifiers were successful only for small samples [34]. The authors selected two subsets of 180 patients with MDD and 180 healthy controls from the BiDirect study. The first subset was analyzed using SVM to identify classifiers for the diagnosis of MDD, and the second subset was used to validate the resulting model. Accuracies ranged from 45.0% to 56.1% for the whole group and from 60.8% to 61.7% for the subgroup with higher depression severity. This resulted in the conclusion that classification models did not translate well in a large realistic population.

## 3. Considerations for ML in Neuroimaging in Depressive Disorders—Treatment Outcomes (Table 2)

### 3.1. Structural Characteristics Related with Depression Treatment Outcomes

The treatment for MDD is determined according to clinical symptoms, and treatments include pharmacotherapy, psychotherapy, and electroconvulsive therapy (ECT). Antidepressants are used as the first-line treatment for depression, and fewer than 50% of patients do not achieve remission [83]. Approximately two-thirds of all patients respond to pharmacotherapy and/or psychotherapy [84], but the remaining one-third are resistant to treatment. The prolonged duration of unremitted MDD increases an individual’s functional loss and overall mental healthcare burden. Therefore, it is very important to predict a patient’s response to particular treatments and to design treatment strategies early at the onset of MDD. Studies have been conducted to identify biomarkers that can predict treatment response, and ML studies are also increasing [9,65,66,67,68,69,70,71,72,85,86].

Gong et al. distinguished between patients with refractory depression and those with non-refractory depression through ML using sMRI data [65]. In this study, the refractory group was defined as MDD patients with a poor response whose Hamilton Depression Rating Scale (HDRS) score did not decrease by more than 50% even after 6 weeks of treatment with two different classes of antidepressants. SVM was applied, and gray matter distinguished between the refractory and non-refractory groups with an accuracy of 69.57%, and white matter distinguished between them with an accuracy of 65.22%. Compared to pre-treatment white matter images, gray matter images showed higher accuracy in predicting the response to antidepressants in patients with MDD. Korgaonkar et al. explored both gray matter volume and FA in 157 patients with MDD, including 103 non-remitters and 54 remitters [66]. Patients received treatment with antidepressants, including escitalopram, sertraline, and venlafaxine, for 8 weeks, and approximately 35% of all participants achieved remission. Using an ML method (decision tree), this study revealed that gray matter volume (smaller left middle frontal gyrus and greater right angular gyrus) and structural connectivity (lower FA values of the left cingulum bundle, right superior fronto-occipital fasciculus, and right superior longitudinal fasciculus) predicted nonremission. It suggested that pre-treatment MRI measures could predict MDD patients who did not respond to antidepressant treatment. Similarly, high accuracy has been reported in ML for predicting treatment response in late-life depression. Patel reported that the optical ADTree model, including measures of structural and functional connectivity, showed an accuracy of 89.47% in a study of 24 patients with depressive disorders (11 responders and 13 non-responders) [9]. A study comparing patients with treatment-refractory depression and healthy controls reported that the patient group and the healthy control group could be discriminated using sMRI, even if MDD patients did not meet the criteria for depressive episodes at the time of MRI scanning [67]. Johnston et al. reported that gray matter reductions in the caudate, insula, and periventricular gray matter supported individual prediction with an accuracy of 85%. Similar to the results of previous sMRI studies [85,86], they suggested an association between reduced volume of the insula and slower recovery/poor prognosis of MDD in ML using sMRI. The result that early treatment cortical thickness (one week into treatment) was more associated with the selective serotonin reuptake inhibitor (SSRI) treatment response than pre-treatment cortical thickness was presented in the Clinical Trial Establishing Moderators and Biosignatures of Antidepressant Response in Clinical Care [68,87]. Bartlett et al. used two methods of random forest (RF) and penalized logistic regression for predicting SSRI treatment response, and psychometric data, demographic data, pre-treatment cortical thickness/volume, and one-week treatment change in cortical thickness/volume were included [68]. RF predicted the remission status more accurately with an accuracy of 63.9%, and they found that frontal lobe structural alterations in the first week of treatment may be associated with long-term treatment efficacy. 

Several ML studies have been reported to predict the response to ECT in MDD [69,70,71]. In a study conducted by Redlich et al., 23 MDD patients received ECT, and they comprised 13 responders and 10 non-responders based on the reduction in their HDRS score (50%) [69]. Structural images obtained before treatment predicted the treatment response with an accuracy of 78.3% (100% sensitivity, 13 of 13 responders). The results of support vector regression (SVR) showed a positive association between predicted and true individual percentages of change in the HDRS score. This study suggests that a higher pre-treatment subgenual cingulate gyrus gray matter volume is associated with a better clinical response. A previous Chinese study using linear kernel SVR also reported that pre-treatment hippocampal subfield volumes predicted whether a patient could achieve remission after ECT and the degree of alleviation of depressive symptoms through the use of ECT [70]. They found that MDD patients with baseline smaller hippocampal subfields had better outcomes, and baseline hippocampal subfield volumes were used to predict the change in depressive symptoms with an overall accuracy of 83.3%. A study was conducted to predict ECT treatment response in a group of patients with depressive disorders and other psychiatric disorders [71,72]. Gärtner et al. predicted the treatment response (percentage of depressive symptom reduction) after ECT in a retrospective study including patients with depressive disorder, bipolar disorder, and schizoaffective disorder [71]. The results showed that the ML method discriminated between responders and non-responders with an accuracy of 69%; gray matter volume in the right parahippocampal gyrus provided the most informative contribution. In a Japanese study, 25 variables along with sMRI data were used as candidate features to predict remission and reduction of depressive symptoms [72]. Compared to the model using only clinical variables, the model including sMRI data showed higher predictability accuracy (70.4% and 92.6%, respectively), and the volumes of the regions including the gyrus rectus, right anterior lateral temporal lobe, cuneus, and third ventricle predicted ECT treatment response. The model including both clinical variables and sMRI data showed the same predictive value as the model using only sMRI data. Previous studies have suggested that pre-treatment sMRI is predictive of ECT treatment, although there are limitations in these studies in that they have a relatively small sample size and include a heterogenous patient group [71,72]. 

A study to predict the improvement of depressive symptoms in adolescents receiving non-pharmacological treatment was also conducted in the United States. Tymofiyeva et al. predicted the treatment response of three months of cognitive behavioral therapy (CBT) using MRI-based structural connectome data [73]. They predicted improvement of depressive symptoms with an accuracy of 83% using J48 classification and right thalamus, left middle frontal gyrus, and baseline depression severity, which were associated with the prediction. Although this study had limitations in that the sample size was small and CBT treatment protocols were heterogeneous, it suggests the possibility of predicting the effect of CBT through brain imaging findings. 

Studies on the search for objective indicators to predict pharmacological and non-pharmacological treatments of depression are increasing, and several ML methods are being used. Current ML studies have suggested the possibility of predicting treatment response through pre-treatment sMRI, but there are some limitations. Most previous studies included relatively small sample sizes and heterogeneous patient groups [71,72]. In addition, classes and dosages of antidepressants have not been strictly determined. Since the pharmacological profile and medication dosage of antidepressants affect brain structure, the effects of these variables cannot be excluded [88]. Furthermore, there is a difference in the prediction accuracy (63–93%) according to the design of the study, and the characteristics of participants and variables associated with predicting the treatment response are inconsistent. The symptoms of depressive disorders are heterogeneous, and the causes of onset are diverse. Multiple types of data, rather than a single type of data, may be helpful in increasing prediction accuracy [89]. To reduce the duration of the untreated period of MDD, high prediction accuracy of the response to each treatment method is essential to plan the treatment strategy. For example, ECT is an effective and well-established treatment for refractory depression and is not generally considered as a first-line treatment in clinical practice for various reasons, such as the potential adverse effect, stigma associated with the treatment, and uncertainty of the treatment mechanism [90]. A more accurate prediction of treatment response may help psychiatrists in clinical decision-making regarding first-line treatment for the management of MDD.

### 3.2. Functional Characteristics Related with Depression Treatment Outcomes

Functional imaging is usually used to identify possible diagnostic markers and refine diagnostic accuracy in patients with depressive disorder, but this is not the only application for this technological advancement. The change in functional connectivity can serve as an indicator for evaluating treatment outcomes and perhaps even the fit between a patient and a certain treatment regimen.

Marquand et al. analyzed task-related fMRI data using the SVM method to examine verbal working memory as a possible biomarker for patients with depression [74]. The brain activity of correlated areas was closer to statistical significance as task difficulty increased, but actual significance was not achieved. Analysis of the treatment response revealed that the most difficult tasks were significantly accurate in predicting the response to 8 weeks of fluoxetine treatment. 

A European research team used an ML strategy called generative embedding, which combines models with classifiers, to predict treatment outcomes in patients with MDD at the single-patient level [75]. Neuroimaging data acquired from the Netherlands Study of Depression and Anxiety were used for supervised learning [91]. The team predicted a given patient’s recovery to be fast or chronic with an accuracy of 79% and fast or gradual with an accuracy of 61%. 

Tian et al. compared the rsFC of 106 patients with MDD and 109 controls to predict treatment outcomes of the antidepressant escitalopram [76]. A linear soft-threshold SVM model discriminated responders from non-responders using a reduction of at least 50% in the HDRS as reference. The anterior cingulate cortex seemed to be the hub for connections for the various interconnections that discriminated responders from non-responders, predicting treatment response with an accuracy of 79.41%.

Liu et al. used an ML technique for model selection in a whole-brain analysis to differentiate MDD patients from healthy controls and further distinguish MDD patients taking amisulpride from those taking placebo to assess the therapeutic effect of dopaminergic enhancement in MDD [77]. The results indicated that the activation and connectivity of reward-related striatal networks were the most predictive, suggesting a possible route by which dopaminergic agents affect treatment outcomes in patients with MDD.

Osuch et al. conducted practical research on the prediction of a medication-class response in patients with mood disorders [78]. A total of 99 subjects (32 with bipolar I disorder, 34 with MDD, and 33 healthy controls) underwent resting-state fMRI, which was used to train a predictive algorithm and construct SVM classifiers. The bipolar disorder group was hypothesized to respond better to mood stabilizers, whereas the MDD group was thought to respond better to antidepressants. This classification resulted in an accuracy of 92.4% in the known-diagnosis group. This method was applied to 12 patients and all had complicated diagnoses. The suggested optimal medication class led to recovery in 11 of 12 cases (approximately 92%).

Hopman et al. predicted the functional connectivity between the left dorsolateral prefrontal cortex (DLPFC) and subgenual cingulate cortex (sgACC) to serve as a biomarker for a repetitive transcranial magnetic stimulation treatment response [79]. Supervised ML analyses of fMRI data from 70 patients with MDD revealed that this was not true. Instead, non-responders showed poor connections between the sgACC and other locations (frontal pole, superior parietal lobule, and occipital cortex) and between the DLPFC and central opercular cortex. These new observations predicted rTMS treatment results with an accuracy of 95.35%.

Cash et al. also investigated possible neuroimaging biomarkers for rTMS treatment outcomes [80]. Data of 47 patients and 29 controls were analyzed using fMRI, resulting in lower activation in the caudate, prefrontal cortex, and thalamic areas in the patient group. Reduced functional connectivity in the default mode and affective networks in patients was also associated with a better treatment response. These features were used to train SVMs, resulting in an rTMS treatment outcome prediction with 85–95% accuracy.

Wang et al. sought to identify biomarkers for the treatment response to ECT, another non-pharmacological treatment option for MDD [81]. They focused on the functional connectivity density (FCD) and rsFC in 23 patients before and after ECT. Neuroimaging data analyses showed that local FCD but not long-range FCD of the left pre-and postcentral gyri and both superior temporal gyri were predictive of changes in HDRS scores after treatment. The SVM-based classification resulted in a prediction accuracy of 72.92%.

Pei et al. combined neuroimaging data with genetic data for more precise modeling of predicting outcomes [82]. The participants were divided into treatment responders and non-responders based on the HDRS score changes after 2 weeks of treatment. Functional connectivity between 14 selected regions of interest and genomic data on selected single nucleotide polymorphisms were acquired. Using SVM with a combination of both datasets resulted in a higher prediction accuracy than when using only one dataset (61% to 86%).

Patel et al. researched patients with late-life depression to find an alternative learning method to the traditional SVM for predicting the diagnosis of and treatment response to depression [9]. They combined various clinical variables with structural and functional neuroimaging data using alternating decision trees. The model showed a diagnostic accuracy of 87.27% using age, mini-mental state examination scores, and structural imaging data as variables, and it showed a treatment response accuracy of 89.47% using structural and functional connectivity data as variables. The best functional connectivity predictors were lower resting-state connections within the dorsal default mode network.

There are many studies trying to find biomarkers for treatment response in MDD patients. In the field of neuroimaging, functional connectivity is a possible candidate since it has resulted in prediction rates with accuracies over 90%, depending on the treatment regimen and the included clinical variables. As an MRI apparatus is for diagnostic purposes, its high costs and low accessibility remain a challenge for its everyday use as a biomarker, but these insights will aid future methods that are more affordable and available.

## 4. Further Considerations in ML for Depressive Disorder

These models may seem close to being clinically applicable. However, it is unknown whether the models can maintain these accuracies when applied to brain images acquired using different scanners and in different populations. In addition, we summarize the issues in the future application of ML to depressive disorders as follows. 

### 4.1. Sample Sizes

When training our model, it was impossible to use information about the entire population. Instead, we could use only a small finite sample. Larger sample sizes are required to use these algorithms. It is quite difficult to acquire sufficient sample data from neuroimaging studies. Small sample sizes and the complexity of the model result in overfitting. In addition, simply increasing the amount of data can worsen overfitting if the number of dimensions also increases. These problems limit the generalizability of the model to clinical settings. Both dimensionality reduction and an increased sample size are required. 

### 4.2. Type of Data including Imaging Modality and Selection of Features from Those Data

No single feature consistently predicted the diagnosis or treatment response across different studies. This reflects the heterogeneity of depression. In addition, neuroimaging data by themselves have limitations in the information they contain. Many features have been hypothesized to be useful for predicting results regarding the diagnosis and treatment outcome. These include sociodemographic, clinical, psychological, neuroimaging, genetic, immune, and endocrine data. It is necessary to include other depression-related clinical variables, such as the clinical characteristics of depression, sex, number of episodes, and multimodal data, as variables for prediction. This further increases the prediction accuracy. As the predictive power of these variables is quite different across different clinical populations, this could limit the generalizability of the study results. In addition, as mentioned for sample sizes, simply increasing the number of features leads to an increase in dimensionality. This also leads to overfitting with respect to the sample size. Researchers must decide how to combine prediction models from different dimensions to achieve accuracy [92].

### 4.3. Training Algorithms and Types of Validation

Many different algorithms have been used, although the most commonly used algorithms are SVM algorithms. Most studies have used supervised prediction and classification algorithms capable of modeling linear and nonlinear relationships to construct predictive models. As there is no one way to reliably integrate all the variables with different modalities into one model, and certain algorithms are more suitable for different combinations of features [93], the proper choice of algorithms and further improvement of algorithms are needed.

### 4.4. Clinical Applicability from Results

The heterogeneity of the samples in studies using ML in relation to clinical characteristics and medication status may limit the generalizability of the results [94]. For example, it is common to use medication-naïve samples in MDD neuroimaging studies. As the effects of medication on neuroimaging findings need to be controlled, many researchers have attempted to compare medication-naïve patients and control groups [7]. MDD is a chronic disease in clinical practice, and many patients suffer from chronic impairments caused by MDD itself. They are also influenced by the medications used to treat MDD. The study results from artificially selected medication-naïve patient groups for comparability issues with healthy controls might be limited in their generalizability and clinical use in real practice, involving a high proportion of chronically depressed patients. The high cost and low availability of neuroimaging facilities in the general population is also another limitation for this field.

Although there are many challenges, it is thought that these ML techniques will eventually integrate the various data to enable individual-level clinical inferences that are applicable to actual clinical practice. This is also expected to be related to personalized precision medicine in the future. 

## Figures and Tables

**Table 1 jpm-12-01403-t001:** Selection of studies investigating machine learning methods for the prediction of diagnosis in depression.

References (Year)	Subjects (Mean Age)	Features	Machine Learning Method	Cross-Validation	Accuracy *	Comments
Foland-Ross et al., 2015 [5]	Baseline 33 adolescents (follow-up: 18 MDD and 15 HC)	Cortical thickness	SVM	stratified 10-fold cross validation	Average accuracy, 69.7%	Girls with an onset of MDD show baseline thinner right medial orbitofrontal cortex and thicker left insula
Kim et al., 2019 [6]	27 HC (15.96 ± 1.02) and 27 MDD (15.48 ± 1.72)	Cortical thickness	SVM	Double LOOCV	94.4% (sensitivity, 92.6% and specificity, 96.3%)	TreeBagging, RF, MLP, AdaBoost, and GBM were used, but they showed lower accuracies than SVM
Qiu et al., 2014 [7]	32 HC (35.0 ± 11.2) and 32 MDD (34.9 ± 11.1)	High-resolution T1-weighted imaging (morphometric parameters)	multivariate SVM	LOOCV	cortical thickness of right hemisphere, 78% (*p* ≤ 0.001)	First-episode, medication-naïve MDD without any psychiatric comorbidities
Qin et al., 2014 [8]	30 HC (35.57 ± 11.73) and 29 MDD (38.97 ± 9.95)	DTI data	SVM with RBF kernel	LOOCV	83.05%	Hubs including the bilateral dorsolateral part of the superior frontal gyrus, the left middle frontal gyrus, the bilateral middle temporal gyrus, and the bilateral inferior temporal gyrus played an important role in diagnosing MDD
Patel et al., 2015 [9]	35 HC and 33 MDD	DTI data, structural imaging, functional imaging	Decision tree	LOOCV	87.3%	The optimal ADTree model selected MMSE score, age, whole brain atrophy, and fluid-attenuated inversion recovery Global WM hyperintensity count for predicting depression diagnosis
Wise et al., 2018 [10]	39 MDD (30.67 ± 8.71) and 8 BPD (29.50 ± 6.21)	High-resolution T1-weighted structural imaging	SVM	LOOCV		Greater gray volume predicted higher MADRS scores
Fung et al., 2015 [11]	19 MDD (30.0 ± 8.9), 16 BPD (26.3 ± 7.9) and HC (27.1 ± 8.4)	T1-weighted structural imaging (Cortical thickness, subcortical volume)	SVM	10-fold cross validation	74.3% (sensitivity, 62.5% and specificity, 84.2%)	Limitation: Effects of medication and chronicity of conditions in BPD and MDD on brain morphological alterations were not estimated
Deng et al., 2018 [12]	36 MDD (29.5 ± 8.6) and 31 BPD (26.3 ± 8.2)	DTI data (FA)	SVM	LOOCV	Left ATR, 68.33% (*p* = 0.018) Right SLF, 66.67% (*p* = 0.029)	RD profile (accuracy) Left CC, 65.57% (*p* = 0.043), Right SLF, 68.25% (*p* = 0.024) Right AF, 72.34% (*p* = 0.008)
Fu et al., 2008 [13]	19 MDD (43.2 ± 8.8) and 19 HC (42.8 ± 6.7)	fMRI data	SVM	LOOCV	86% (sensitivity 84% and specificity 89%)	Lateral temporal cortex, amygdala, and visual processing networks contributed most
Cao et al., 2014 [14]	39 MDD (27.99 ± 7.49) and 37 HC (28.22 ± 6.47)	fMRI data	SVM	LOOCV	84%	Inferior orbitofrontal, supramarginal gyrus, inferior parietal lobule-posterior cingulated gyrus, and middle temporal gyrus-inferior temporal gyrus contributed most
Mourao-Miranda et al., 2011 [15]	19 MDD (43.2 ± 8.8) and 19 HC (42.8 ± 6.7)	fMRI data	SVM	Nested LOOCV	52%	Patients were identified as outliers during facial recognition, with 30% of outliers responding to antidepressants, whereas 89% of non-outliers responded
Zeng et al., 2012 [16]	24 MDD (31.83 ± 10.99) and 29 HC (33.62 ± 10.29)	fMRI data	SVM	LOOCV	94.3%	550 discriminating functional connections; 100% accuracy for patients, 89.7% for controls
Guo et al., 2018 [17]	59 MDD and 31 HC, 29 MDD and 24 HC	fMRI data	SVM	LOOCV	92.22% and 90.57%	Voxel-mirrored homotopic connectivity (VMHC) alterations examined for two separate samples
Wei et al., 2013 [18]	20 MDD (34.3 ± 8.2 and 20 HC (30.8 ± 8.7)	fMRI data	SVM	LOOCV	90% (sensitivity 95% and specificity 85%)	Right fronto-parietal and default mode networks showed deficits, while the left fronto-parietal, ventromedial prefrontal, and salience network were excess networks
He et al., 2021 [19]	40 MDD (40.05 ± 12.32) and 34 HC (34.44 ± 11.76)	fMRI data, peripheral blood	SVM	LOOCV	85.1%	MicroRNA-9, thought to be a neural substrate of childhood maltreatment, integrated into analysis
Ramasubbu et al., 2016 [20]	45 MDD (37 ± 11) and 19 HC (33 ± 10)	fMRI data	SVM	5-fold cross validation	66%	Patients grouped by severity. Mild to moderate (58%) and severe (52%) groups showed lower accuracies
Ramasubbu et al., 2019 [21]	22 MDD (27.36 ± 7.5) and 22 HC (28.09 ± 2.71)	fMRI data	SVM	Nested LOOCV	77.3% (sensitivity 75% and specificity 80%)	Arterial spin labeling MRI was used to measure cerebral blood flow (CBF). Regional CBF of cortical, limbic, and paralimbic regions contributed to classification.
Yamasita et al., 2020 [22]	149 MDD and 564 HC from four sites, 185 MDD and 264 HC from five sites	fMRI data	LASSO	Nested cross validation	70%	Functional connectivity differences were identified in multisite data, which were applied for classification on another multisite dataset for validation.
Nouretdinov et al., 2011 [23]	19 MDD and 19 HC	fMRI data	TCP	Conformal prediction	89.5% and 92.1% at 90% confidence	Two sad-face recognition tasks used to classify patients using the TCP method; prediction accuracy at least 90% at 90% confidence level
Hahn et al., 2011 [24]	30 patients (MDD, BPD) and 30 HC	fMRI data	GP classification	LOOCV	60%	Sad face, happy face, anxious face, neutral face, anticipation of no reward, anticipation of large reward, anticipation of no loss, and avoiding small loss were significant classifiers
Rosa et al., 2015 [25]	30 patients (MDD, BPD) and 30 HC	fMRI data by Hahn et al. (2011)	Linear L1-norm regularized SVM	Nested cross validation	85%	A novel sparse network based discriminative modeling framework was applied on existing data. Higher accuracies were reached
Shi et al., 2021 [26]	92 MDD, 460 MDD, and 470 HC	fMRI data	Relevance vector regression, eXtreme Gradient Boosting classification	LOOCV, 10-fold cross validation	86.3%	Gray matter density and fractional amplitude of low-frequency fluctuation predicted sleep disturbance in patients. The model was applied to a multicenter dataset for validation.
Guo et al., 2017 [27]	38 MDD (28.4 ± 9.68) and 28 HC (26.6 ± 9.4)	fMRI data	Multikernel SVM		97.54%	A method generating a high order minimum spanning tree functional connectivity network was used to reduce computing consumption and produce a scale conducive to subsequent network analysis
Sato et al., 2015 [28]	25 MDD and 21 HC	fMRI data	Maximum entropy linear discriminant analysis	LOOCV	78.3% (sensitivity 72.0%, specificity 85.7%)	Guilt selective connections used for classification
Han et al., 2019 [29]	25 MDD and 21 schizophrenia	fMRI data	Nonnegative matrix factorization	LOOCV	82.6%	“Triple network” (default mode, salience, central executive) used to distinguish MDD patients from schizophrenia patients
Yu et al., 2013 [30]	19 MDD (26.65 ± 7.62), 32 schizophrenia (24 ± 5.66), and 38 HC (24.44 ± 4.45)	fMRI data	SVM	LOOCV	80.9% (84.2% for MDD, 81.3% for schizophrenia, 78.9% for HC)	Altered connections in medial prefrontal, anterior cingulate, thalamus, hippocampus, and cerebellum for both patient groups; differences in prefrontal, amygdala, and temporal poles
Grotegerd et al., 2013 [31]	10 MDD (36.8 ± 10.1) and 100 BPD (36.8 ± 8.5)	fMRI data	SVM	LOOCV	90%	Medial prefrontal, orbitofrontal regions contributed to classifying unipolar and bipolar depression
He et al., 2020 [32]	63 MDD (35.35 ± 11.02) and 63 HC (31.78 ± 10.56)	fMRI data	SVR	LOOCV		Left and right amygdala/hippocampus predicted trait sadness; medical prefrontal/anterior cingulate and amygdala/parahippocampal gyrus predicted state anhedonia scores
Maglanoc et al., 2020 [33]	170 MDD (38.7 ± 13.3) and 71 HC (41.8 ± 13.1)	fMRI data	Shrinkage discriminant analysis	10-fold cross validation		Low model performance for classification of depression or anxiety symptoms
Sundermann et al., 2017 [34]	Two subsets of 180 MDD and 180 HC	fMRI data	SVM	LOOCV	56.1%	The subgroup with a higher symptom severity showed a higher classification accuracy (61.7%).

TreeBagging, tree-based bagging; RF, random forest; MLP, multilayer perception; AdaBoost, adaptive boosting; GBM, gradient boosting machine; SVM, support vector machine; SVR, support vector regression; RBF, Gaussian radial basis; ADTree, alternating decision tree; LASSO, least absolute shrinkage and selection operator; TCP, transductive conformal predictor; GP, Gaussian process; LOOCV, leave-one-out cross-validation; FA, fractional anisotropy; RD, radial diffusivity; GM. gray matter; WM. white matter; ATR, anterior thalamic radiation; SLF, superior longitudinal fasciculus; AF, arcuate fasciculus; CC, cingulum cingulate; MADRS, Montgomery–Asberg Depression Rating Scale; HDRS, Hamilton Depression Rating Scale; MDD, major depressive disorder; BPD, bipolar disorder; HC, healthy controls; Accuracy *, highest accuracies presented.

**Table 2 jpm-12-01403-t002:** Selection of studies investigating machine learning methods for the prediction of treatment outcomes in depression.

References (Year)	Subjects (Mean Age)	Features	Machine Learning Method	Cross-Validation	Accuracy *	Comments
Patel et al., 2015 [9]	11 MDD responders and 13 MDD non-responders	DTI data, structural imaging, functional imaging	Decision tree	LOOCV	89.5%	The optimal ADTree model selected MMSE score, age, whole brain atrophy, and fluid-attenuated inversion recovery. Global WM hyperintensity count for predicting depression diagnosis
Gong et al., 2011 [65]	22 non-refractory MDD (39.17 ± 12.88) and 23 refractory MDD (40.43 ± 12.58)	GM and WM	SVM	LOOCV	69.6% (GM) and 65.22% (WM)	Participants were treated with one of three classes of antidepressants: tricyclic, serotonin–norepinephrine reuptake inhibitor, and selective serotonin reuptake inhibitor
Korgaonkar et al., 2015 [66]	54 remitted MDD and 103 non-remitted MDD	GM volume and DTI data (FA)	Decision tree	Hold-out	85.0% (GM volume) and 84.0% (FA)	Participants were randomized to receive flexibly-dosed escitalopram, sertraline, or venlafaxine-ER for 8 weeks
Johnston et al., 2015 [67]	20 treatment-refractory MDD (51.80 ± 11.23) and 21 HC (46.14 ± 13.97)	T1-weighted brain imaging (GM)	SVM	LOOCV	85% (sensitivity, 85% and specificity, 86%)	MDD participants had experienced lifetime and/or current chronic episodes of depression, not necessarily meeting criteria for MDD at time of scanning
Bartlett et al., 2018 [68]	63 remitters (34.59 ± 12.23) and 121 non-remitters (38.40 ± 13.69)	T1-weighted brain imaging (cortical thickness)	RF, PLR	10 repetitions of 5-fold cross-validation	63.9% (sensitivity, 22.6% and specificity, 85.8%)	Patients with early onset MDD (before age 30) and chronic (episode duration >2 years) or recurrent MDD (≥2 recurrences) were enrolled. Remission status was predicted more accurately with RF than PLR
Redlich et al., 2016 [69]	23 ECT-treated MDD (45.7 ± 9.8), with 13 responders and 10 non-responders	High-resolution T1-weighted structural imaging (GM volume)	SVM, SVR	LOOCV	78.3% (sensitivity, 100% and specificity, 50%)	Brief-pulse ECT was conducted three times a week with antidepressants (mean number of sessions, 14)
Cao et al., 2018 [70]	24 severe MDD (31.3 ± 10.8), with 12 remitters and 12 non-remitters	T1-weighted structural imaging (GM volume)	SVR	LOOCV	Overall, 83.3% (sensitivity, 91.7% and specificity, 75%)	All the patients were under severe unipolar depression and received eight sessions of modified ECT
Gaertner et al., 2021 [71]	39 responders (50.23 ± 17.53) and 32 non-responders (51.31 ± 18.09)	Structural MRI	SVM with a linear kernel	LOOCV	69% (sensitivity, 67% and specificity, 72%)	Schizoaffective disorder (4%) and BD (13%) were included. Twelve sessions of ECT were administered, and patients with partial response had extra ECT-sessions (mean no. sessions: 13.61 ± 4.34)
Takamiya et al., 2020 [72]	20 remitters and seven non-remitters	High-resolution T1-weighted structural imaging (GM volume) and clinical variables	SVM, SVR	LOOCV	90% (sensitivity, 100% and specificity, 71%)	Clinical variables included age, sex, diagnosis, psychotic features, family history of mood disorder, duration of episode, illness duration, previous ECT, and the score of each item of HDRS-17
Tymofiyeva et al., 2019 [73]	30 MDD (16.0 ± 1.3)	DTI data	Decision tree (J48)	10-fold cross validation	83% (sensitivity, 82% and specificity, 84%)	All patients underwent CBT, and six patients received antidepressants with CBT; 19 improvers and 11 non-improvers were included
Marquand et al., 2008 [74]	20 MDD (43.7 ± 8.6) and 20 HC (43.7 ± 8.3)	fMRI data	SVM	LOOCV		Statistical significance for response prediction not achieved
Frassle et al., 2020 [75]	85 MDD	fMRI data	SVM	LOOCV	79% (chronic vs. fast remission), 61% (gradual improvement vs. fast remission)	Data from the Netherlands Study of Depression and Anxiety were used to classify chronic patients, gradual improvement, and fast remission
Tian et al., 2020 [76]	106 MDD and 109 HC	fMRI data	SVM	LOOCV	79.4%	Multicenter data analyzed while assuming an HDRS score reduction of at least 50% as response after escitalopram monotherapy
Liu et al., 2020 [77]	57 MDD (31 amisulpride, 26 placebo) and 28 HC	fMRI data	Elastic net regularization	Nested cross validation	77% (MDD vs. HC), 59% (amisulpride vs. placebo)	Striatal network functional connectivity changes were most predictive for classification, suggesting a dopaminergic role in treatment outcome
Osuch et al., 2018 [78]	34 MDD (19.7 ± 2.6), 32 BPD (21.3 ± 2.9), and 33 HC (20.2 ± 2.0)	fMRI data	SVM	Nested cross validation	92.4% (MDD vs. BPD), 92% (medication class response prediction)	Diagnostic classification also succeeded in predicting the optimal medication class of response, where BPD patients responded to mood stabilizers, and MDD patients responded better to antidepressants.
Hopman et al., 2021 [79]	70 MDD (41.93 ± 11.67)	fMRI data	SVM	5-fold cross validation	95.35%	Medication resistant patients were treated with rTMS and analyzed to predict short term and long-term treatment response. Sustained response was associated with stronger anterior cingulate/occipital cortex connectivity
Cash et al., 2019 [80]	47 MDD (43 ± 12) and 29 HC (39 ± 15)	fMRI data	SVM	LOOCV	85~95%	Reduced connectivity in default mode and affective network was associated with better rTMS response
Wang et al., 2018 [81]	23 MDD (38.74 ± 11.02) and 25 HC (39.52 ± 8.07)	fMRI data	SVM	LOOCV	72.92%	Local functional connectivity density of left pre/postcentral gyri, both superior temporal gyri were predictive of ECT treatment response
Pei et al., 2020 [82]	98 MDD	fMRI data, venous blood	SVM	LOOCV	86%	fMRI data were combined with genetic data on selected single nucleotide polymorphisms for classification of responders and non-responders to medication, resulting in higher accuracy than fMRI data alone (61%)

RF, random forest; SVM, support vector machine; SVR, support vector regression; PLR, penalized logistic regression; ADTree, alternating decision tree; LOOCV, leave-one-out cross-validation; FA, fractional anisotropy; GM. gray matter; WM. white matter; MADRS, Montgomery–Asberg Depression Rating Scale; ECT, electroconvulsive therapy; HDRS, Hamilton Depression Rating Scale; rTMS, repeated transcranial magnetic stimulation; MDD, major depressive disorder; BPD, bipolar disorder; HC, healthy controls. Accuracy *, highest accuracies presented.

## Data Availability

Not applicable.

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
