# Peer review of "Personalized Diagnosis and Treatment for Neuroimaging in Depressive Disorders"

_jpm, 2022, doi:10.3390/jpm12091403_

Round 1
Reviewer 1 Report
In the review article, the authors used machine learning techniques to summarize the publications of neuroimaging studies in depressive disorders, in the aspects of diagnosis of subtypes of major depressive disorders, and the prediction of treatment responses. This comprehensive and holistic review can give readers a current status of the neuroimaging technique applied to depressive disorders.
Here are some suggestions from the reviewer:
1. In line 225, Zeng et al. analyzed resting-state functional connectivity in patients with what diagnosis?
2. The author should write a brief introduction of LIBSVM and cite a reference. (line 237)
3.In the 4.4 Clinical applicability from results,
Another limitation is that imaging facilities are expensive and usually not equipped in private clinics.
4. The use of abbreviations should be adequate. Abbreviations shown in the abstract should be separated from the main text. For example, MDD (line 28), machine learning (ML) (line70), LIBSVM (line 237), HDRS (line 395). Please check if there are others.
Reviewer 2 Report
This is the review of the manuscript: Personalized Medicine for Neuroimaging in Depressive Disorders
Authors have included 94 references in this review, presenting current ML related methods for advancing diagnoses accuracy and treatment efficacy. There are several major issues with this review manuscript, which need to be addressed prior to publication.
Major issues:
- Most important issue: There are two writing styles in this article, one is for structural imaging (2.1, 3.1) another is for functional imaging (2.2. 3.2). It’s also happening in the introduction section, where author listed several reason why fMRI can be a useful tool from line61-64, but simply says “sMRI has been conventionally used in patients with clinical depression ” in line 60 to describe sMRI. This is very odd since it’s one review, not two.
- Title is not directly related to the content. When author use “personalized medicine” in the title, there are very limited references in the article related to medicine. It’s better to change the title into “Personalized diagnosis and treatment for neuroimaging in Depressive Disorders.”
- There are many repetitive sentences in the introduction, which should be re-phrased in a shorter version. There’s zero reference in the introduction section, which is very odd too.
- The logic in introduction section is not clear. For example, from line 29-30: “indicates that the characteristics of MDD biological findings 29 might not be confined to a few parameters but may also affect multiple dimensions of 30 data, including neuroimaging data”. If author believe that neuroimaging can’t be the only factor in diagnosis accuracy and treatment optimization, why they support the idea that with ML, bio imaging can, again, be the solo factor in the entire article?
- From line 38-42, “However, MDD diagnosis has several limitations since the diagnostic process is time-consuming and depends on the subjective judgement of the clinician. In clinical practice, it is common not to receive additional help from brain imaging 40 although brain imaging provides a noninvasive evaluation of brain structure and function 41 and provides a deeper understanding of the neuropathophysiology of MDD.” Why author said there are several limitations, but only list two: time-consuming and subjective judgement? Does ML can help to overcome these two factors so that imaging can be taken into consideration in the future clinical practice?
- What’s the biomarkers for current clinical diagnosis and treatment design?
- There are many repetitive sentences from line 80 to line 89.
- When the structural imaging sections (2.1, 3.1) used a comprehensive style to review the references, functional imaging sections (2.2, 3.2) simply list each reference one by one through the entire section. Either one is not the best option. Please try to integrate the references in a more systematic manner.
Please addresses these questions for the first round of review.
